# OpenReview forum: "Systematic Rectification of Language Models via Dead-end Analysis"
_ICLR.cc/2023/Conference — ICLR 2023 poster_

### Official Review · Reviewer_ZwPr · 2022-10-21

**Confidence:** 2
**Correctness:** 4
**Technical Novelty And Significance:** 3
**Empirical Novelty And Significance:** 3
**Recommendation:** 8

**Clarity, Quality, Novelty And Reproducibility:**

I am not expert enough in these topics to provide a relevant evaluation of the quality, soundness and novelty of the method.
Nonetheless, I found the paper very well written, easy to follow and to understand and convincing.
Many details are provided in the appendices, which should help reproduce the results.

Minor formatting comments:

"methods like Raffel et al. (2020);" --> like those proposed by
"computationally expensive and slow Gehman et al. (2020); Yang & Klein (2021)." --> citation formatting

**Strength And Weaknesses:**

Strength: The method is based on a theory that looks sound, and is confirmed by the results. The proposed model can work for any LLM, alleviating the need for retraining. The paper is well written and could be understood by someone (like me) not very familiar with LLMs and RL. The results look impressive.

Weakness: The TestFilter method seems to be simpler and to work quite well already. The best results presented in this paper are in combination with TestFilter. While showing that the two methods can be complementary, it might be interesting to cary out more comparisons between the proposed method and TestFilter, in particular for the human evaluation and the qualitative results.

**Summary Of The Paper:**

This paper presents a method to prevent existing large language models to produce toxic discourse. Contrary to existing methods that modify the training procedure for LLMs or eliminate tokens at inference based on rules, the proposed method uses reinforcement learning to compute, at each step of the generation, a probability that the final discourse will be toxic and modifies the sampling probability accordingly.

The authors propose a separate small rectification model that may be applied to any LLM as long as they have the same vocabulary, based on the $\beta$-dead-end theory in RL. They compare the new method with retraining-based and decoding-based baselines and report competitive results. The method can be combined with the TestFilter decoding-based method to out-perform existing techniques.

**Summary Of The Review:**

This paper is clear and well written. It addresses a relevant and important problem. The method looks soundly grounded in theory and the results are good. As detailed in the other sections, more analysis and comparisons could be useful to evaluate the method.

---

> ### Author Response · Authors · 2022-11-13
> **Response**
>
> Thank you for your comments and positive feedback! We added a short discussion in the Appendix.
>
> **1. “It might be interesting to carry out more comparisons between the proposed method and TestFilter”**
>
> We appreciate the suggestion. We should first highlight that the test time filtering baseline is a Monte Carlo method and even if it may seem to work nicely in some examples, there is no guarantee that the final generations are non-toxic, except when N is very large, which can be intractable. In direct contrast, for our method, we have a theoretical guarantee.
>
> Second, TestFilter is less effective when the input prompt is extremely toxic. The paper shows the evaluation results on the challenging subset of RealToxicityPrompts. For GPT-3, TestFilter achieves 17% toxicity probability, and our method achieves 4.9%. Moreover, in terms of memory usage, the TestFilter baseline requires at least N times more memory, which can be expensive when the generated sequence is long.
>
> Finally, we would like to reiterate what the reviewer also touched upon: combining our method with Monte Carlo sampling (like TestFilter) has three advantages: 1) improving detoxification, even more, 2) maintaining our formal guarantees, and 3) not requiring too many samples.

---

### Official Review · Reviewer_6y1n · 2022-10-21

**Confidence:** 3
**Clarity, Quality, Novelty And Reproducibility:** see above
**Correctness:** 3
**Technical Novelty And Significance:** 3
**Empirical Novelty And Significance:** 3
**Recommendation:** 6

**Strength And Weaknesses:**

The task of making LMs output less harmful language is interesting and
important. The authors present connections to the RL literature,
including some new theorems about about rectification/dead ends/etc.
The authors should be commended for their willingness to empirically
engage with larger language models than I've seen in prior RL+NLP
work, e.g., GPT-2 XL (1.5B) and GPT-3 (175B). Their choice of
baselines are reasonable (save for my biggest concerns, detailed
below). I tend to agree with the authors that developing methods
assuming only specific levels of API access will be important
in the regime of very large language models, and so efforts in that
domain are appreciated.


My biggest concern are missing empirical comparisons to work that
explores a very similar setup --- i.e., using RL from human feedback
to "detoxify" an LM using
perspective/realtoxicityprompts. Specifically, the authors mention
that the two main approaches to detoxification in this setting are
retraining-based/decoding-based. But, there are works that have
readily applied RL methods to this setup before, e.g., Lu et al. 2022
improve upon the improved decoding time method for detoxification of
Liu et al. 2021 in a comparable setup.

- Liu, Alisa, et al. "DExperts: Decoding-time controlled text
  generation with experts and anti-experts." arXiv preprint
  arXiv:2105.03023 (2021).

- Lu, Ximing, et al. "Quark: Controllable Text Generation with
  Reinforced Unlearning." arXiv preprint arXiv:2205.13636 (2022).

It's difficult to directly compare the results from Lu et al. 2022
with those that are presented here. (my best effort:  the
author's gpt2 baseline is 61% toxic and 18 perplexity, and the most
aggressive detox is 10% toxic and 22 perplexity. Lu et al.'s gpt2
baseline is 52% toxic and 11 perplexity (on wikitext). Their method
gives 4% toxic and 12 perplexity (on the in-domain training data),
which seems better (?).) Both methods conduct human evaluations,
but are not directly comparable.

Overall, it's difficult to know which method performs best at this
task because the comparison is missing --- and because the Lu et
al. method has been out for a while, I think it's fair to expect some
comparison with it. That being said, it appears that the Lu et
al. method may be solving a slightly different problem: instead of
updating the parameters of the policy, the proposed rectification is
(at inference time) a decoding-based method. Thus, some APIs, e.g.,
GPT-3, may be more compatible. Though, notably --- GPT-3 may be
fine-tuned as well, but not via an API that requires access to
model-internal representations.  But, if that's the case, I would have
appreciated a comparison to Liu et al. 2021 instead, which also
appears to be a decoding-time method for this "detoxification" task.

A few additional thoughts:

- For a decoding method, I would have liked to have seen a reward
  filtering baseline, i.e., sample N times from the model, and select
  the single sample with lowest model-estimated perplexity.

- The human evaluation results are not that definitive --- ignoring
  ties, the author's model wins in 63% of cases versus the baseline in
  terms of detoxifying.

- I would have appreciated a more thorough discussion of the potential
  for reward hacking in detoxification. This is briefly alluded to in
  Appendix E, but there are missing references, e.g., about the
  shortcomings of perspective, for example:

Hosseini, Hossein, et al. "Deceiving google's perspective api built
for detecting toxic comments." arXiv preprint arXiv:1702.08138 (2017).

**Summary Of The Paper:**

The authors frame the task of LM "detoxification" as an RL problem.
They propose "rectification," which requires only access only to token
probabilities from an LM API, rather than internal states.  The method
down-weights tokens that are likely to cause eventual toxic discourse,
treating toxic generations probabilistically as "dead ends".
According to both human and automatic evaluation, the author's new
method provides new pareto-optimal trade-offs between fluency and
toxicity compared to retraining/decoding baselines.

**Summary Of The Review:**

Overall, the paper offers a strong empirical contribution to RL tuning
of LMs from human feedback in the setting where policy parameters are
not allowed to be updated. The idea of training a policy to do
decoding time modification of logits is quite interesting, and it's
promising that the method can be applied to models which provide a
more limited API like GPT-3. The authors make it clear in the abstract
that they believe this setting is useful, but --- it appears there's
some missing discussion (and possibly empirical comparison) to methods
which use RL from human feedback (with full policy access) or
decoding-time methods beyond DAPT.


=== after the response period:

The authors added more experiments that I had requested which indeed better situate Rect in comparison to the prior work I had mentioned. Thank you! I still am somewhat concerned about the human evaluation results being not as definitive --- even if they are statistically significant, the magnitude is relatively small. Nonetheless, I have boosted my score.

---

> ### Author Response · Authors · 2022-11-13
> **Response**
>
> Thank you for your thorough and detailed comments. We added the two baselines in our manuscript. We conducted a new set of experiments to compare our approach with the two baselines in Appendix D.
>
> **1. “missing empirical comparisons to work that explores a very similar setup --- i.e., using RL from human feedback to "detoxify" an LM”**
>
> - Lu, Ximing, et al. "Quark: Controllable Text Generation with Reinforced Unlearning."
>
>     Thank you for pointing this out! We believe this work is related to our work, but not directly comparable (we cited Lu et al. in the related work section). An important assumption of our work is that we can only obtain token probabilities from LM, rather than internal states. Lu et al. method updates the parameters of the LM (i.e., retraining-based method). While fine-tuning large models like GPT-3 might be possible, it is still extremely computationally expensive (let alone the fact that many such models are not open-sourced and only the owner company is able to do so). The proposed approach avoids the situation where we need to fine-tune the underlying LM whenever the dataset changes.
>
>     Those being said, there are also three significant differences between Lu et al.’s setting and ours.
>     - First, Lu et al. 2022 evaluate their method on 10K non-toxic prompts from the RealToxicityPrompts dataset. The numbers cited in the review (i.e., 10% toxic) are evaluation results on the challenging subset of the RealToxicityPrompts dataset. In our paper, we present the non-toxic evaluation results in the non-toxic columns in Table 1 (columns 5 & 8).
>     - Second, Lu et al. 2022 use 85% of RealToxicityPrompts as the training set. In our work, however, we follow Gehman et al., 2020 and only use RealToxicityPrompts as the evaluation set (out-of-domain training data).
>     - Third, Lu et al. 2022 use PerspectiveAPI as their reward function. In our experiments, we trained a BERT-classifier as our reward function to prevent the RL agent from overexploiting the API.
>
>     In summary, our setting is different and more difficult compared with Lu et al. 2022’s setting. For comparison, we add evaluation results following the same setup as Lu et al. 2022 (85k in-domain training data & PerspectiveAPI as a reward function) in Appendix D. On the same test set (10k non-toxic prompts), the following are the new evaluation results:
>
>     | Model | exp. max. | prob. | dist-2 | dist-3 |
>     | ----------- | ----------- |  ----------- |  ----------- |  ----------- |
>     | GPT-2 | 0.527 | 52.0% | 0.85 | 0.85 |
>     | Quark | 0.196 | 3.5% | 0.80 | 0.84 |
>     | PPLM | 0.520 | 51.8% | 0.86 | 0.86 |
>     | Dexpert | 0.314 | 12.8% | 0.84 | 0.84 |
>     | GeDi | 0.363 | 21.7% | 0.84 | 0.83 |
>     | Rect $(\epsilon=0.2)$ | 0.227 | 3.4% | 0.85 | 0.86 |
>     | Rect $(\epsilon=0.4)$ | 0.184 | 1.2% | 0.86 | 0.87 |
>
> - Liu, Alisa, et al. "DExperts: Decoding-time controlled text generation with experts and anti-experts."
>
>     We agree that this work is relevant and should be compared with our method (both decoding-based methods). On 10k non-toxic prompts, Liu et al.’s method achieve 0.314 average maximum toxicity and 12.8% toxicity probability. Our method achieves 0.184 average maximum toxicity and 1.2% toxicity probability, and both are noticeably better than the results of Liu et al. We added these comparisons to Table 8 in Appendix D.
>
> **2. “For a decoding method, I would have liked to have seen a reward filtering baseline, i.e., sample N times from the model, and select the single sample with lowest model-estimated perplexity.”**
>
> We are not sure if we understand the reviewer's question. Would you please confirm if you mean perplexity or toxicity? (If perplexity, the baseline doesn’t make logical sense to us. If toxicity, we already have this baseline in the paper (i.e., Test-time Filtering).)
>
> **3. “The human evaluation results are not that definitive --- ignoring ties, the author's model wins in 63% of cases versus the baseline in terms of detoxifying.”**
>
> We run a two-tailed sign test and the p-value p = 0.0438, which is smaller than the significance level of 0.05. Thus, the null hypothesis is rejected and the human evaluation results are statistically significant.
>
> **4. “I would have appreciated a more thorough discussion of the potential for reward hacking in detoxification. This is briefly alluded to in Appendix E, but there are missing references, e.g., about the shortcomings of perspective”**
>
> We didn’t completely understand what the reviewer meant by “reward hacking”. Our understanding is that reward hacking refers to the RL agent learning to abuse some possibly unknown “shortcuts” in the PerspectiveAPI rather than actually performing detoxification (is our understanding true?). To avoid this, we have trained a BERT-based toxicity classifier as the reward function for RL training. We also performed the manual evaluation and qualitative analysis of the model’s outputs to demonstrate the effectiveness of our approach.

---

> > ### Comment · Reviewer_6y1n · 2022-11-13
> > **thanks for the reply!**
> >
> > Hi there --- thanks for the reply. I will respond more fully later to everything, but just following up on a few quick clarifications:
> >
> > # select the single sample with lowest model-estimated perplexity
> >
> > you're right --- this should say "best reward" rather than "lowest model-estimated perplexity". I'll take a second look at your baseline.
> >
> > # reward hacking
> >
> > yes, that's what I meant by reward hacking. Thanks for the response, and I will take a second look.

---

> > ### Comment · Reviewer_6y1n · 2022-11-29
> > **Thanks for your response!**
> >
> > Hi there --- I had a chance to more fully read your response and will take it into account in ongoing reviewer discussions about your submission. Specifically, thanks for running additional empirical comparisons --- it looks like Rect is quite promising and requires even less API access compared to prior methods. I still am somewhat concerned about the human evaluation results being not as definitive --- even if they are statistically significant, the magnitude is relatively small. I will update my review (+ score) in accordance with this new information.

---

### Official Review · Reviewer_p8Zr · 2022-10-27

**Confidence:** 3
**Correctness:** 4
**Technical Novelty And Significance:** 3
**Empirical Novelty And Significance:** 3
**Recommendation:** 6

**Clarity, Quality, Novelty And Reproducibility:**

The paper is well organized and written in general. It is novel to formulate the detoxification problem as a reinforcement learning problem, and the authors clearly present the theory and the proof of the related theorems.

**Strength And Weaknesses:**

Strength:

(1) It is novel to contemplate a language generation procedure as a standard Markov decision process and formulate the detoxification problem as a reinforcement learning problem.

(2) One learned model can be used to detoxify many language models if they share the same vocabulary.

(3) The experimental results show that the proposed approach can generate better results compared to the baselines in detoxification by both human and automatic evaluation.

Weaknesses:

(1) Although the detoxification model would be smaller than the pre-trained language models, it is unclear that whether the proposed method has advantage in computational efficiency over existing detoxification methods both at the training and inference phases.

(2) I doubt the rectification method would decrease the diversity of generated texts. It would be better to add some experiments to evaluate the generated contexts in their diversity.


**Summary Of The Paper:**

This work proposes a detoxification method that aims to reduce the risk of large language models (LLMs) generating toxic contexts such as rude, disrespectful, and unreasonable language without retraining the LLMs. The idea behind this method is to reduce the probability of selecting the next token if that token will cause a toxic text to be generated eventually with some level of certainty. The authors take a langue generation procedure as a standard Markov decision process and formulate the detoxification tasks as an auxiliary reinforcement learning problem. They demonstrate that the proposed method can significantly mitigate toxicity by both human and automatic evaluation.

**Summary Of The Review:**

This word presents a detoxification method by extending dead-end theory from the recent reinforcement learning methods to alleviate toxicity for large language models. A dead-end state was defined to be a state from which a toxic text would be generated with a high probability. The main idea is to avoid toxic texts to be generate by adjusting the probability of token selection proportional to the risk of ultimate toxicity when the generation process is finished. This study was is motivated and timely. The experimental results show that the proposed method can significantly mitigate toxicity by both human and automatic evaluation. However, the proposed method might decrease the diversity of generated texts. Some experiments are required to evaluate the generated contexts in their diversity.

---

> ### Author Response · Authors · 2022-11-13
> **Response**
>
> Thank you for your comments and feedback. We have added diversity evaluation results in Appendix D in the manuscript.
>
>
> **1. “Although the detoxification model would be smaller than the pre-trained language models, it is unclear that whether the proposed method has advantage in computational efficiency over existing detoxification methods both at the training and inference phases.”**
>
> - First, for large language models (e.g., gpt-3, gpt-2 XL), the proposed approach has a significant advantage in terms of computational efficiency compared to retraining-based baselines. Our method doesn’t require updating the LM parameters, and the detoxification model can be significantly smaller than the LM.
> - Compared with PPLM, a decoding-based method, our method is about 40 times faster at inference time since PPLM requires at least two forward passes and one backward pass at each decoding step. This can be expensive when the LM is large.
> - The self-debiasing (SD) method requires an additional input with a self-debiasing template. Therefore, the decoding time is at least doubled. For our method, because of the small size of the detoxification model, the increased decoding time is smaller than the decoding time of the original LM.
> - The test filer baseline needs to generate up to N times as many continuations in the worst case, and which often happens when the input prompt is toxic. In contrast, our model does not require that and only alters the way that LM generates tokens at each step.
>
> **2. “I doubt the rectification method would decrease the diversity of generated texts. It would be better to add some experiments to evaluate the generated contexts in their diversity.”**
>
> We appreciate the suggestion. We added diversity evaluation results in Appendix D. The results show that the proposed method doesn’t decrease the diversity of generated text.

---

### Decision · Program_Chairs · 2023-01-20

**Decision:**

Accept: poster

**Justification For Why Not Higher Score:**

Decent contribution, but limited to specific sub-community.

**Justification For Why Not Lower Score:**

The technical contributions are simple, novel and effective empirically. The findings will be useful in informing future work in this area.

**Metareview: Summary, Strengths And Weaknesses:**

This paper proposes a method called rectification to reduce toxic generation in large language models (LLMs). The method requires access to only token probabilities of an LM and does not require any fine-tuning. This is done by treating the generation problem as an MDP and providing rewards based on whether the eventual generation is toxic, treating them as dead-ends and downweighting intermediate generated tokens accordingly. The method is empirically verified to provide improved tradeoffs between fluency and toxicity compared to several baselines.
The reviewers appreciated the novelty of the setup and effectiveness of the method in experiments, had concerns around the diversity of generations and comparison to additional baseline, both of which were addressed in the author response. I encourage the authors to incorporate those results into the final version of the main paper (esp. the Lu et al. baseline which is currently added to appendix D).

**Note From Pc:**

if the above contains the word "oral" or "spotlight" please see: "oral" presentation means -> notable-top-5% and "spotlight" means -> notable-top-25%. As stated in our emails, we are disassociating presentation type from AC recommendations